# Timing of the reticular lamina and basilar membrane vibration in living gerbil cochleae

Wenxuan He[1], David Kemp[2], Tianying Ren[1]*

[1]Oregon Hearing Research Center, Department of Otolaryngology, Oregon Health & Science University, Portland, United States; [2]University College London Ear Institute, University College London, London, United Kingdom

**Abstract** Auditory sensory outer hair cells are thought to amplify sound-induced basilar membrane vibration through a feedback mechanism to enhance hearing sensitivity. For optimal amplification, the outer hair cell-generated force must act on the basilar membrane at an appropriate time at every cycle. However, the temporal relationship between the outer hair cell-driven reticular lamina vibration and the basilar membrane vibration remains unclear. By measuring sub-nanometer vibrations directly from outer hair cells using a custom-built heterodyne low-coherence interferometer, we demonstrate in living gerbil cochleae that the reticular lamina vibration occurs after, not before, the basilar membrane vibration. Both tone- and click-induced responses indicate that the reticular lamina and basilar membrane vibrate in opposite directions at the cochlear base and they oscillate in phase near the best-frequency location. Our results suggest that outer hair cells enhance hearing sensitivity through a global hydromechanical mechanism, rather than through a local mechanical feedback as commonly supposed.
DOI: https://doi.org/10.7554/eLife.37625.001

*For correspondence:
rent@ohsu.edu

Competing interests: The authors declare that no competing interests exist.

## Introduction

The exceptional sensitivity of mammalian hearing has been attributed to a micromechanical feedback system inside the cochlea, also called 'the cochlear amplifier' or 'cochlear active process' (*Dallos et al., 2008*; *Davis, 1983*; *Fettiplace and Hackney, 2006*; *Hudspeth, 2014*; *Robles and Ruggero, 2001*; *Russell et al., 2007*). When sound-induced basilar membrane vibrations deflect hair bundles of the outer hair cells, mechanoelectrical transduction of these cells generates the receptor potential (*Dallos et al., 1982*; *Russell and Sellick, 1983*). In response to the membrane potential change, mammalian outer hair cells change their length and generate force primarily through the somatic motility driven by the motor protein, prestin (*Ashmore, 2008*; *Brownell et al., 1985*; *Liberman et al., 2002*; *Mammano and Ashmore, 1993*; *Mellado Lagarde et al., 2008*; *Ren et al., 2016a*; *Santos-Sacchi, 1989*; *Zheng et al., 2000*). This cellular force is thought to be directly applied to the basilar membrane at its generation location on a cycle-by-cycle basis, consequently amplifying the sound-induced basilar membrane vibration and boosting hearing sensitivity (*Dallos et al., 2008*; *de Boer, 1995b*; *Dong and Olson, 2013*; *Hudspeth, 2014*; *Liu and Neely, 2009*; *Reichenbach and Hudspeth, 2014*).

For optimal amplification, the cellular force must act on the basilar membrane at an appropriate time at every vibration cycle (*Dallos et al., 2008*; *Nilsen and Russell, 1999*). Therefore, timing of the cochlear feedback has been a central research topic in the field of auditory neuroscience since the cochlear amplifier was proposed (*Ashmore, 2008*; *Davis, 1983*; *Gold, 1948*; *Gummer et al., 1996*). Because cochlear amplification depends on normal cochlear metabolism and the integrity of cochlear mechanical properties (*Cooper and Rhode, 1992*; *Fisher et al., 2012*; *Lee et al., 2015*;

**eLife digest** What is the quietest sound the ear can detect? All sounds begin as vibrating air molecules, which enter the ear and cause the eardrum to vibrate. We can detect vibrations that move the eardrum by a distance of less than one picometer. That's one thousandth of a nanometer, or about 100 times smaller than a hydrogen atom. But how does the ear achieve this level of sensitivity?

Vibrations of the eardrum cause three small bones within the middle ear to vibrate. The vibrations then spread to the cochlea, a fluid-filled spiral structure in the inner ear. Tiny hair cells lining the cochlea move as a result of the vibrations. There are two types of hair cells: inner and outer. Outer hair cells amplify the vibrations. It is this amplification that enables us to detect such small movements of the eardrum. Inner hair cells then convert the amplified vibrations into electrical signals, which travel via the auditory nerve to the brain.

The bases of outer hair cells are connected to a structure called the basilar membrane, while their tops are anchored to a structure called the reticular lamina. It was generally assumed that outer hair cells amplify vibrations of the basilar membrane via a local positive feedback mechanism that requires the hair cells to vibrate first. But by comparing the timing of reticular lamina and basilar membrane vibrations in gerbils, He et al. show that this is not the case. Outer hair cells vibrate after the basilar membrane, not before. This indicates that outer hair cells use a mechanism other than commonly assumed local feedback to amplify sounds.

The results presented by He et al. change our understanding of how the cochlea works, and may help bioengineers to design better hearing aids and cochlea implants. Millions of patients worldwide who suffer from hearing loss may ultimately stand to benefit.

DOI: https://doi.org/10.7554/eLife.37625.002

Lee et al., 2016; Nuttall et al., 1991; Ren and Nuttall, 2001; Rhode, 1971; Robles and Ruggero, 2001; Ruggero and Rich, 1991; van der Heijden and Versteegh, 2015), the cochlear feedback has to be investigated ultimately in living cochleae. Timing of cochlear feedback was studied in vivo by measuring basilar membrane vibrations at different locations across the width of the basilar membrane in guinea pig (Nilsen and Russell, 1999). Based on their observation, the authors suggest that forces generated by the outer hair cells directly drive the region of the basilar membrane beneath the Deiters' cells. The temporal relationship between the reticular lamina and basilar membrane vibration was previously observed in guinea pigs using a time-domain optical coherence tomography system as a homodyne interferometer (Chen et al., 2011). It was reported that the phase of the reticular lamina vibration led the phase of the basilar membrane vibration at the best frequency (Figure 5, Chen et al., 2011). This phase lead has been thought to ensure the right timing of the outer hair cell force for cochlear amplification. However, the mouse micromechanical data measured using a heterodyne low-coherence interferometer showed no significant phase difference between the reticular lamina and basilar membrane vibration at the best frequency (Ren et al., 2016b). This discrepancy may have been caused by the animal species and technical differences, that is mouse versus guinea pig and homodyne interferometry versus heterodyne interferometry. Although a number of studies have been conducted recently to measure micromechanical responses in living cochleae (Gao et al., 2014; Lee et al., 2016; Ramamoorthy et al., 2016; Recio-Spinoso and Oghalai, 2017; Cooper et al., 2018) the timing of the cochlear feedback remains unclear. Since the apical end of outer hair cells is directly connected to the reticular lamina, the reticular lamina vibration can reflect the movement of the outer hair cell under physiological conditions. Therefore, the timing of the cochlear feedback was determined in this study by measuring the latency difference between the reticular lamina and basilar membrane vibration using a custom-built heterodyne low-coherence interferometer (Hong and Freeman, 2006; Ren et al., 2016a; Ren et al., 2016b). The present data collected from the gerbil, one of the most commonly used animals for auditory research, demonstrate for the first time that the reticular lamina vibrates after, not before, the basilar membrane vibration.

# Results

## Reticular lamina and basilar membrane vibrations in sensitive gerbil cochleae

A representative data set from one of twenty-three sensitive cochleae is presented in *Figure 1*. The displacement of the reticular lamina response to 30 dB SPL tones (0 dB SPL = 20 µPa) increased and then decreased with frequency, forming a sharp peak at ~26 kHz (best frequency, BF) (*Figure 1A*). While displacements increased proportionally with sound level at frequencies < 15 kHz, they increased at a much smaller rate near the best frequency. The response peak became broader and shifted toward low frequencies as the sound level increased. In contrast to sharp tuning at 30 dB SPL, the displacement curve at 80 dB SPL showed no response peak. Displacements of the basilar membrane at 30 and 40 dB SPL were ~10 fold smaller than those of the reticular lamina not only near the best frequency but also at lower frequencies (*Figure 1B*). Basilar membrane response also reached the maximum at ~26 kHz as did the reticular lamina. For ~333 fold sound level increase

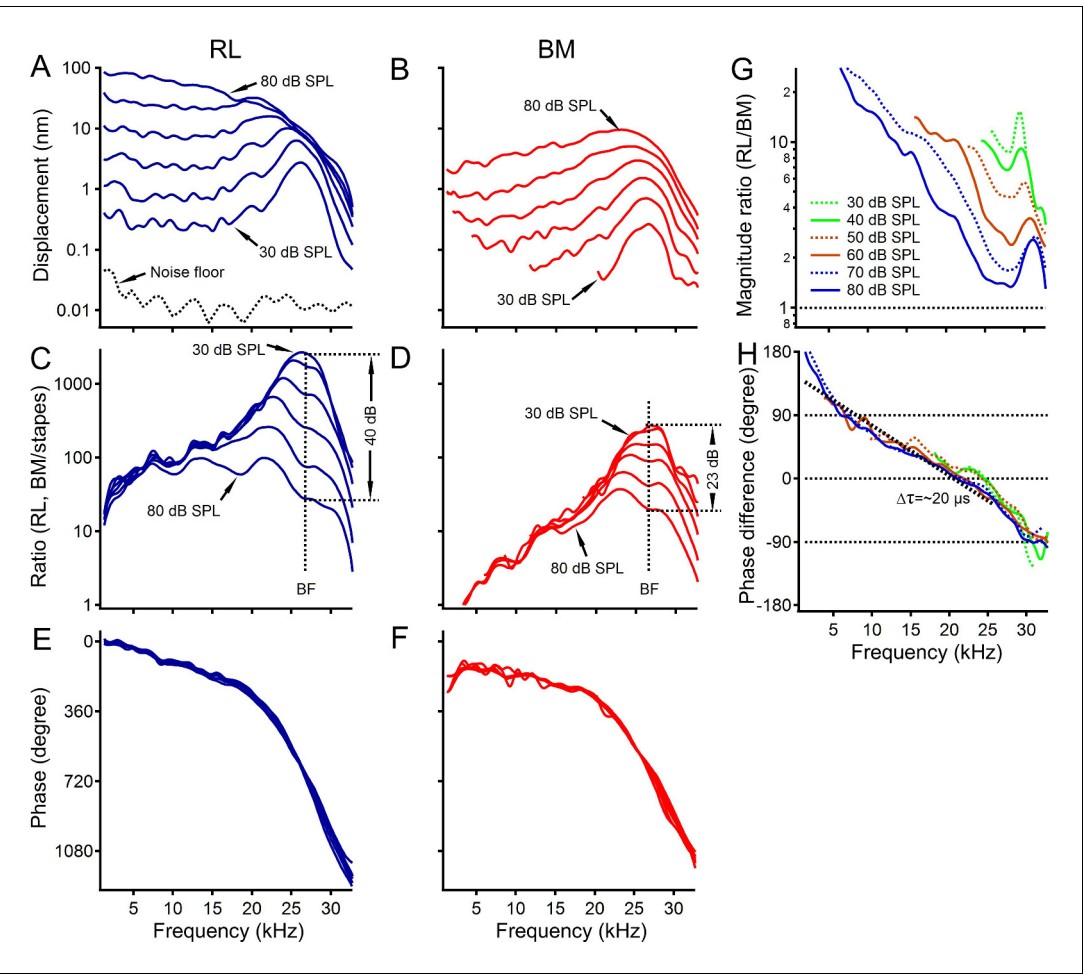

**Figure 1.** The reticular lamina and basilar membrane vibration in a sensitive gerbil cochlea. (**A, B**) Displacements of the reticular lamina (RL) and basilar membrane (BM) as a function of frequency at different sound levels. The noise floor is indicated by the black dotted line in panel A. (**C, D**) Ratios of RL and BM displacements to stapes displacements at different sound levels. (**E, F**) RL and BM phase as a function of frequency. (**G**) The ratios of the RL displacement to the BM displacement as a function of frequency at different sound levels. (**H**) Phase difference between the RL and BM vibration. Line types and colors in this panel are the same as those in plot (**G**). The slope of the linear regression line (thick black dotted) indicates that the latency of the RL vibration is ~20 µs greater than that of the BM vibration at 70 dB SPL.

DOI: https://doi.org/10.7554/eLife.37625.003

from 30 to 80 dB SPL, the displacement of the basilar membrane at ~26 kHz increased by ~23 fold compared with only ~3.3 fold increase of the reticular lamina displacement at the same frequency. These differences were confirmed by the displacement ratios of the reticular lamina (*Figure 1C*) and basilar membrane (*Figure 1D*) vibration to the stapes vibration. The reticular lamina showed ~40 dB nonlinear compression near the best frequency, which is ~17 dB greater than that of the basilar membrane (~23 dB). Phase responses of the reticular lamina (*Figure 1E*) are similar to those of the basilar membrane (*Figure 1F*) except for a slightly steeper phase slope at frequencies < 15 kHz. Thus, highly sensitive, sharply tuned nonlinear responses of the reticular lamina and basilar membrane in the gerbil (*Figure 1A–D*) are similar to those measured from the basal turn of mouse cochleae under sensitive conditions (*Ren et al., 2016b*).

The magnitude relationship between the reticular lamina and basilar membrane vibration is presented in *Figure 1G* by the ratio of the reticular lamina displacement to the basilar membrane displacement as a function of frequency. All displacement ratios are greater than one, indicating that reticular lamina vibrations were greater than those of the basilar membrane at different frequencies and sound levels. Near the best frequency (~26 kHz), the displacement ratio was ~10 at 30 dB SPL, decreasing with sound level and becoming ~1.5 at 80 dB SPL. In contrast to well documented sharply tuned basilar membrane vibration at the best frequency (*Robles and Ruggero, 2001*), the greatest displacement ratios were observed at frequencies far below the best frequency and at high sound levels.

The temporal relationship between the reticular lamina and basilar membrane vibration was determined by phase difference as a function of frequency and is presented in *Figure 1H*. At frequencies < 10 kHz, reticular lamina phase led basilar membrane phase by up to 180 degrees. This phase lead decreased with frequency and the phase difference became slightly negative at the best frequency. The slope of the linear regression line (thick black dotted) of the phase difference curve at 70 dB SPL indicates that the latency of the reticular lamina is ~20 µs greater than that of the basilar membrane.

## Reticular lamina and basilar membrane vibrations in insensitive gerbil cochleae

Compared to sensitive responses (red lines in *Figure 2A*), displacements of the reticular lamina vibration decreased dramatically at all frequencies under postmortem conditions (blue lines in *Figure 2A*), while the basilar membrane vibration decreased only near the best frequency (blues lines in *Figure 2B*). In contrast to the sharp peak of 30 dB SPL sensitive responses at ~26 kHz, the broad peaks of reticular lamina and basilar membrane postmortem responses shifted toward low frequencies (blue lines in *Figure 2A,B*). Although the basilar membrane vibration at frequencies < 20 kHz did not decrease significantly, the reticular lamina vibration decreased by >10 fold over the same frequency range. This is unexpected since cochlear amplification has been believed to be effective only near the best frequency (*Robles and Ruggero, 2001*). The overlapping blue curves in *Figure 2C,D* and the equally separated blue curves in *Figure 2A,B* indicate linear growth for postmortem responses. In contrast to the lack of significant change in basilar membrane phase (*Figure 2F*), the reticular lamina phase decreased by up to 180 degrees at frequencies < 10 kHz (*Figure 2E*). The displacement ratio of the reticular lamina to basilar membrane vibration decreased dramatically at all frequencies (compare blue to red lines in *Figure 2G*). The frequency-dependent phase lead (red lines in *Figure 2H*) was absent under postmortem conditions (blue lines in *Figure 2H*). Thus, postmortem data in *Figure 2* demonstrate that the magnitude and phase differences between the reticular lamina and basilar membrane vibration depend on normal cochlear metabolism. While the reticular lamina and basilar membrane vibration in the gerbil (*Figure 1*) are qualitatively similar to those in the mouse (*Figure 1C–J*, *Ren et al., 2016b*), there are the following quantitatively differences. Although the measurements were taken from similar longitudinal locations in the basal turn of the cochlea in both species, the best frequencies of the reticular lamina and basilar membrane vibration in the gerbil are significantly lower than those in the mouse. Compared to the mouse postmortem data (*Figure 2G and H*, *Ren et al., 2016b*), the basilar membrane vibrated more than the reticular lamina in postmortem gerbil cochleae (*Figure 2*).

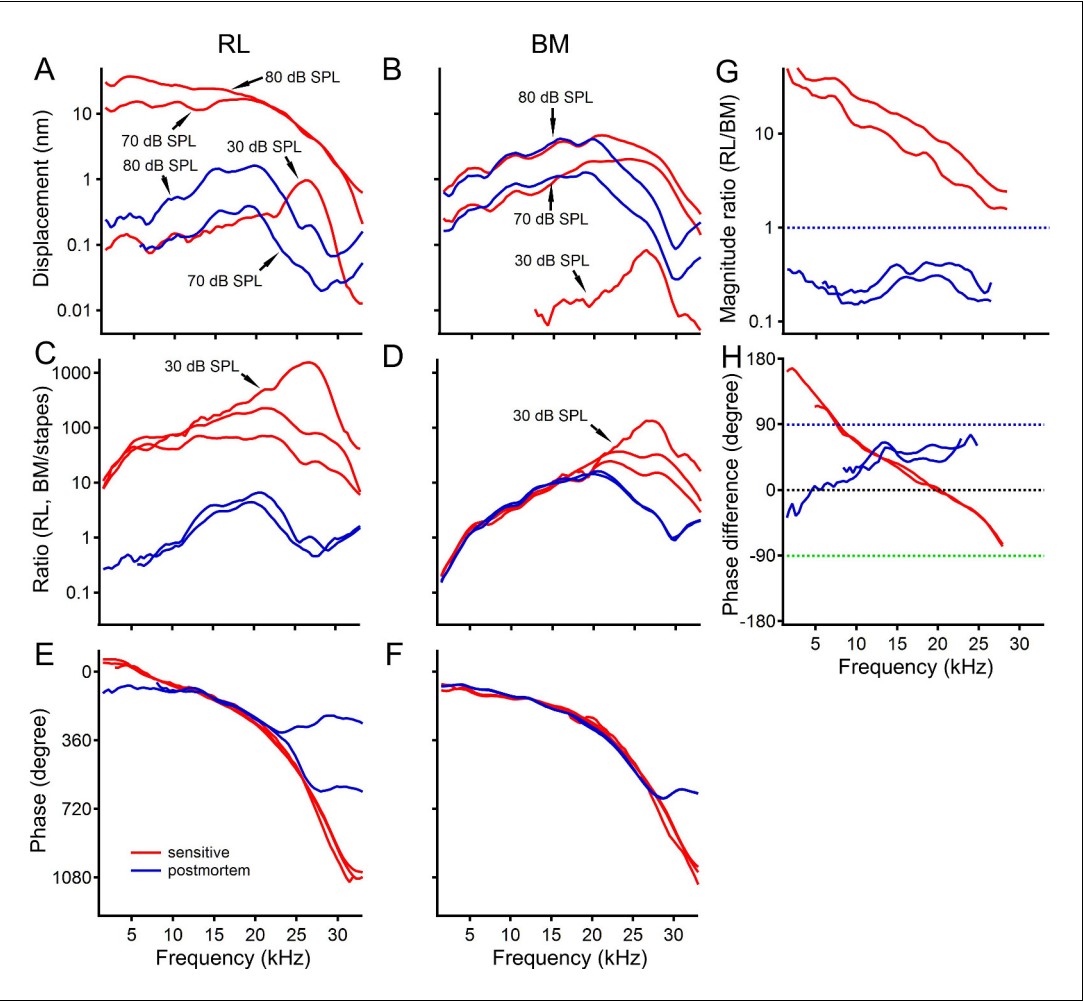

**Figure 2.** Postmortem changes in reticular lamina and basilar membrane vibrations. (**A**) Postmortem reticular lamina (RL) displacements (blue lines) are significantly smaller than those under sensitive conditions (red lines) at all frequencies. (**B**) Basilar membrane (BM) displacements deceased only near the best frequency (~26 kHz) under postmortem conditions (blue lines). (**C, D**) Sensitive (red lines) and insensitive (blue lines) RL and BM magnitude transfer functions. (**E**) RL phase decreased by up to 180 degrees at low frequencies under postmortem conditions. (**F**) No significant difference between sensitive and insensitive BM phase curves. (**G**) The ratio of RL displacement to BM displacement decreased dramatically at all frequencies under postmortem conditions (blue lines). (**H**) The phase difference (~180 degrees) at low frequencies (red lines) is absent in the postmortem cochlea (blue curves).
DOI: https://doi.org/10.7554/eLife.37625.004

## Latency difference between the tone-induced reticular lamina and basilar membrane vibration

To determine the latency difference, the phase and magnitude differences between the reticular lamina and basilar membrane vibration were measured as a function of frequency at different sound levels in seven sensitive gerbil cochleae (*Figure 3*). For statistical analysis, frequency axes were normalized to the best frequency for each animal. The displacement ratio of the reticular lamina vibration to the basilar membrane vibration was the largest at the low-frequency end and decreased with frequency reaching the lowest level near the best frequency (*Figure 3A,C,E,G*). The averaged phase data show that the reticular lamina phase led the basilar membrane phase by >135 degrees at low frequencies. This phase lead decreased with frequency and approached zero near the best frequency (*Figure 3B,D,F,H*). While the magnitude ratio decreased with the sound level near the best frequency (*Figure 3I*), the phase difference showed no significant change at the same frequency (*Figure 3J*). Consequently, the latency differences derived from the slope of the phase difference

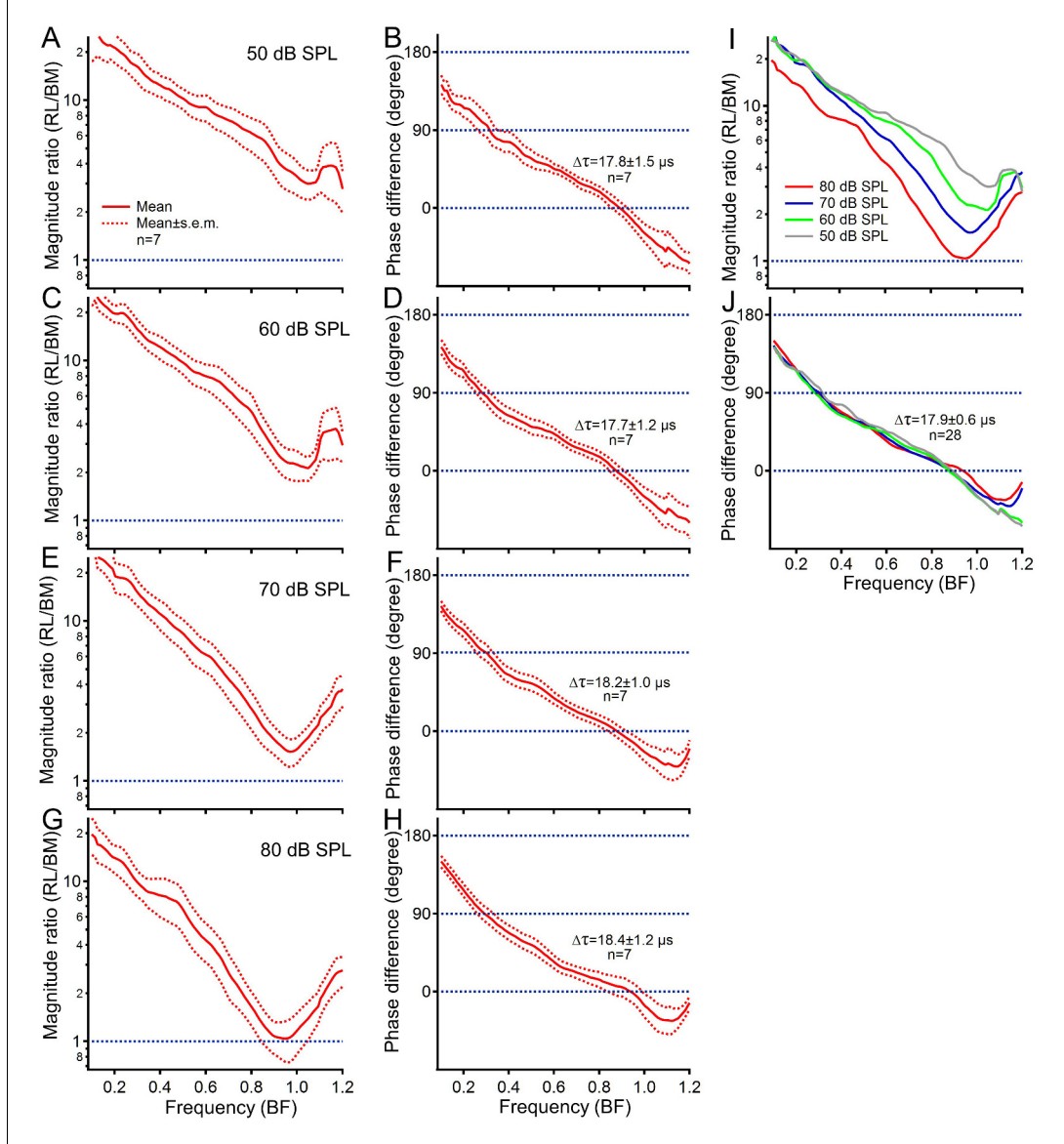

**Figure 3.** Phase and latency differences between the reticular lamina and basilar membrane vibration across animals. (A, C, E, G) Means and standard errors of the displacement ratio of the reticular lamina (RL) to basilar membrane (BM) at 50, 60, 70, and 80 dB SPL. (B, D, F, H) Phase differences between the RL and BM vibration. Latency (Δτ) was calculated from the phase slope and presented next to phase difference curves. (I) The magnitude ratio decreases with sound level near the best frequency. (J) Phase difference as a function of frequency shows no significant change with sound level. Data in panels (A–H) are presented as mean ± SEM.

DOI: https://doi.org/10.7554/eLife.37625.005

curves show no significant change across stimulus levels (ANOVA test: F = 0.037, p=0.990). This allows us to calculate the latency difference between the reticular lamina and basilar membrane vibration across animals and sound levels. The grouped data from seven gerbils at four different sound levels demonstrate that the latency of the reticular lamina vibration is ~17.9 μs greater than that of the basilar membrane (17.9 ± 0.6 μs, n = 28) (*Figure 3J*). To compare this result with that in mice, the latency difference in sensitive mouse cochleae was derived from recently published phase data (*Figure 3F*, *Ren et al., 2016b*) using the same experimental procedures as those in the current study. The latency difference between the reticular lamina and basilar membrane vibration in mice (12.1 ± 0.6 μs, n = 5) is significantly smaller than that in gerbils (17.9 ± 0.6 μs, n = 28) (t = 7.775, p<0.01, n = 33). The smaller latency difference in mice likely results from higher best frequencies.

## Latency difference between the click-induced reticular lamina and basilar membrane vibration

To confirm latency difference revealed by phase data in *Figure 3*, the reticular lamina and basilar membrane response to clicks were measured by recording instantaneous displacement as a function of time at different sound levels. In the sensitive living cochlea, a 10-μs rarefaction click caused a large displacement of the reticular lamina toward the scala tympani indicated by the first negative peak (red curves in *Figure 4A*) at ~0.40 ms, while the basilar membrane moved toward the scala vestibuli (indicated by the first positive peak of blue curves in *Figure 4A*). In addition to the opposite directions, the peak magnitude of the reticular lamina displacement is ~eight fold larger than that of the basilar membrane vibration, which increased proportionally with the sound level. Following the initial opposite movements, both the reticular lamina and basilar membrane oscillated periodically and the vibration magnitudes decreased gradually approaching to their equilibrium positions. To clearly show the temporal relationship, the displacements of the reticular lamina and basilar membrane vibration at 90 dB-p (0 dB-p = 20 μPa of peak sound pressure) were plotted as a function of time with different magnitude scales in *Figure 4C*. Since the initial peaks indicate the maximum displacements of the reticular lamina and basilar membrane vibration, which effectively stimulate the cochlea, the initial peak times $T_A$ and $T_B$ (*Figure 4C*) are used to present the latencies of the basilar membrane and reticular lamina response respectively, and the latency difference was determined by $T_B$-$T_A$. Latency difference measured at 90 dB-p in ten sensitive cochleae (32.6 ± 1.5 μs, n = 10) demonstrates that the outer hair cell-driven reticular lamina vibration occurs after the basilar membrane vibration. This latency difference is greater than that derived from the phase data in *Figure 3*, likely due to the stimulus difference, that is tone verse click. Despite their initial opposite movements, the reticular lamina and basilar membrane moved synchronously in the same direction at ~0.51 ms ($T_C$) (*Figure 4C*), because the phase change of the reticular lamina vibration with time is slower than that of the basilar membrane. Similarly, the first period of the reticular lamina ($T_{RL}$) is greater than that of the basilar membrane ($T_{BM}$) (the low panel in *Figure 4A*) indicating that the starting frequency of the reticular lamina is lower than that of the basilar membrane vibration. The magnitude of the reticular lamina vibration decreased dramatically under postmortem conditions and became comparable to that of the basilar membrane (*Figure 4B,D*). Moreover, the direction of the first peak of the reticular lamina displacement changed from negative to positive, that is from toward the scala tympani to toward the scala vestibuli, which became consistent with the direction of the basilar membrane vibration. These changes indicate that the reticular lamina moves passively following the basilar membrane vibration under postmortem conditions. The data at sound levels below 70 dB SPL were not shown because the initial peak of the basilar membrane vibration was too small to be reliably detected. Thus, the time-domain data in *Figure 4* confirm that the reticular lamina vibrates after, not before, the basilar membrane vibration. Moreover, the initially opposite displacements and following synchronous movements of the reticular lamina and the basilar membrane in *Figure 4A and C* are consistent with the ~180° phase difference at low frequencies and in-phase vibrations at the best frequency in *Figure 3*.

## Discussion

This paper reports the first in vivo measurement of the latency difference between the outer hair cell-driven reticular lamina vibration and the basilar membrane vibration. The present data demonstrate that the latency of the reticular lamina vibration is greater than that of the basilar membrane vibration, and there is no significant phase difference between the two structures near the best frequencies. This result is consistent with the mouse data measured using heterodyne interferometry (*Ren et al., 2016b*) but inconsistent with the guinea pig data, which showed that the phase of the reticular lamina vibration leads the phase of the basilar membrane vibration by ~90° at the best frequency and the phase lead decreases with sound pressure level (*Chen et al., 2011*). The current result also confirms recent studies in the mouse (*Ren et al., 2016b*) and in the guinea pig (*Recio-Spinoso and Oghalai, 2017*) that the physiologically vulnerable reticular lamina vibration is significantly greater than the basilar membrane vibration not only at the best frequency but also at low frequencies. Since an ~90° phase lead of the reticular lamina vibration is thought to be required for cochlear feedback to amplify basilar membrane vibration (*Chen et al., 2011*; *Gummer et al., 1996*; *Nilsen and Russell, 1999*; *Robles and Ruggero, 2001*; *Russell and Nilsen, 1997*), and since

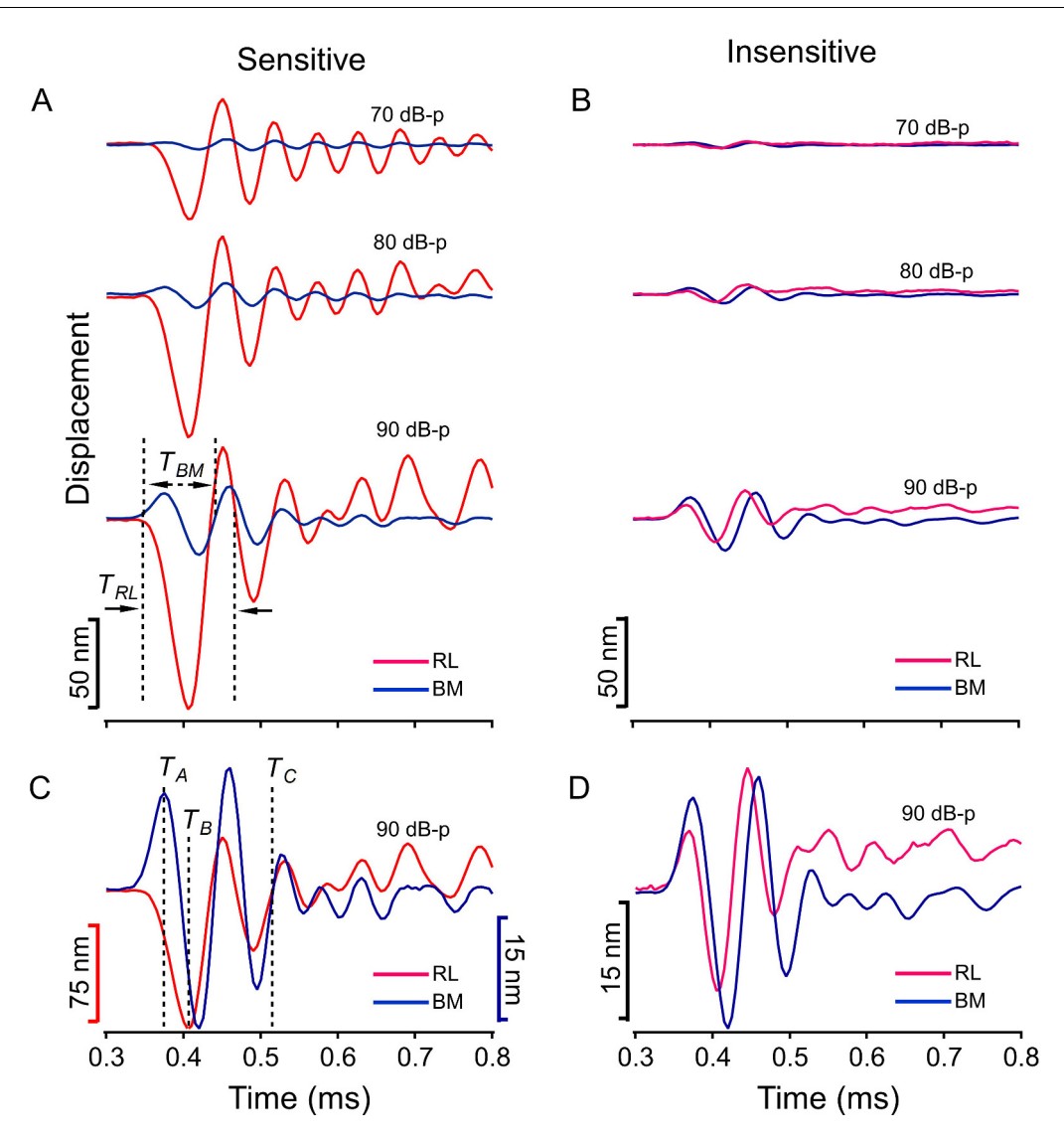

**Figure 4.** Time waveforms of the reticular lamina and basilar membrane vibration. (**A**) Displacements of the reticular lamina (red) and basilar membrane (blue) vibration in response to 10-µs rarefaction clicks as a function of time at 70, 80, and 90 dB-p (0 dB-p = 20 µPa of the peak sound pressure). (**B**) The reticular lamina and basilar membrane response to clicks measured under postmortem conditions. (**C**) To show the temporal relationship, the reticular lamina and basilar membrane response at 90 dB-p are plotted with different magnitude scales. (**D**) Postmortem responses of the reticular lamina and basilar membrane at 90 dB-p. $T_A$, the arriving time of the first peak of the basilar membrane vibration; $T_B$, the arriving time of the first peak of the reticular lamina vibration; $T_C$, the time when the reticular lamina and basilar membrane vibration become in phase; $T_{BM}$ and $T_{RL}$, the first periods of the basilar membrane and reticular lamina vibration.

DOI: https://doi.org/10.7554/eLife.37625.006

cochlear amplification has been predicted to work only near the best frequency location (*de Boer, 1995a*; *Dong and Olson, 2013*; *Liu et al., 2017*; *Liu and Neely, 2009*; *Meaud and Grosh, 2012*; *Motallebzadeh et al., 2018*; *Ni et al., 2016132016*; *Ramamoorthy et al., 2007*; *Wang et al., 2016*), the current result is inconsistent with the local cochlear feedback hypothesis. Instead, the latency difference between the reticular lamina and basilar membrane vibration found in this study supports the global hydromechanical mechanism for cochlear amplification (*Ren et al., 2016b*), which is discussed below.

The longitudinal patterns of the reticular lamina and basilar membrane vibrations at the best-frequency (26 kHz) are presented by plotting displacements and phases (*Figure 1A,B,E,F*) as a function of the location along the cochlear length (blue and red lines in *Figure 5A–D*), which was derived from the stimulus frequency according to the cochlear frequency-location function (*Müller, 1996*). While a 30 dB SPL tone-induced response occurred at a < 0.2 mm region at the best-frequency location (*Figure 5A,C*), a 70 dB SPL tone-induced vibration extended from the best-frequency location to the base (*Figure 5B,D*). The overlapping phase curves near the 2.05 mm location (blue and red lines in *Figure 5C,D*) indicate that the reticular lamina and basilar membrane vibrated approximately in the same direction at the best-frequency location. The ~180 degree separation between the blue and red lines near the cochlear base (*Figure 5D*) indicates opposite vibrations of the reticular lamina and basilar membrane.

The outer hair cell-driven active movement was estimated by vector subtraction of the basilar membrane vibration from the measured reticular lamina vibration and is presented by green lines in *Figure 5A–D*. The overlapping blue and green lines in *Figure 5A and C* indicate that, at low sound levels, reticular lamina vibration is dominated by outer hair cell-driven movement. At 70 dB SPL, the outer hair cell-driven responses saturated near the best-frequency location, indicated by the diverged blue and green lines near the response peak (*Figure 5B*).

Time waveforms of the outer hair cell-driven reticular lamina vibrations (green) and the basilar membrane vibrations (red) in *Figure 5E,F* were derived from magnitude and phase in *Figure 5A–D* at two sequential times. The time difference between solid and dotted green curves is ~6 µs, equivalent to ~57 degree phase difference at 26 kHz. For clearer comparison, basilar membrane time waveforms (red curves) were shifted down by 4 nm in *Figure 5E* and by 50 nm in *Figure 5F* respectively. When the basilar membrane moves upward to the scala vestibuli near the cochlear base (red arrow

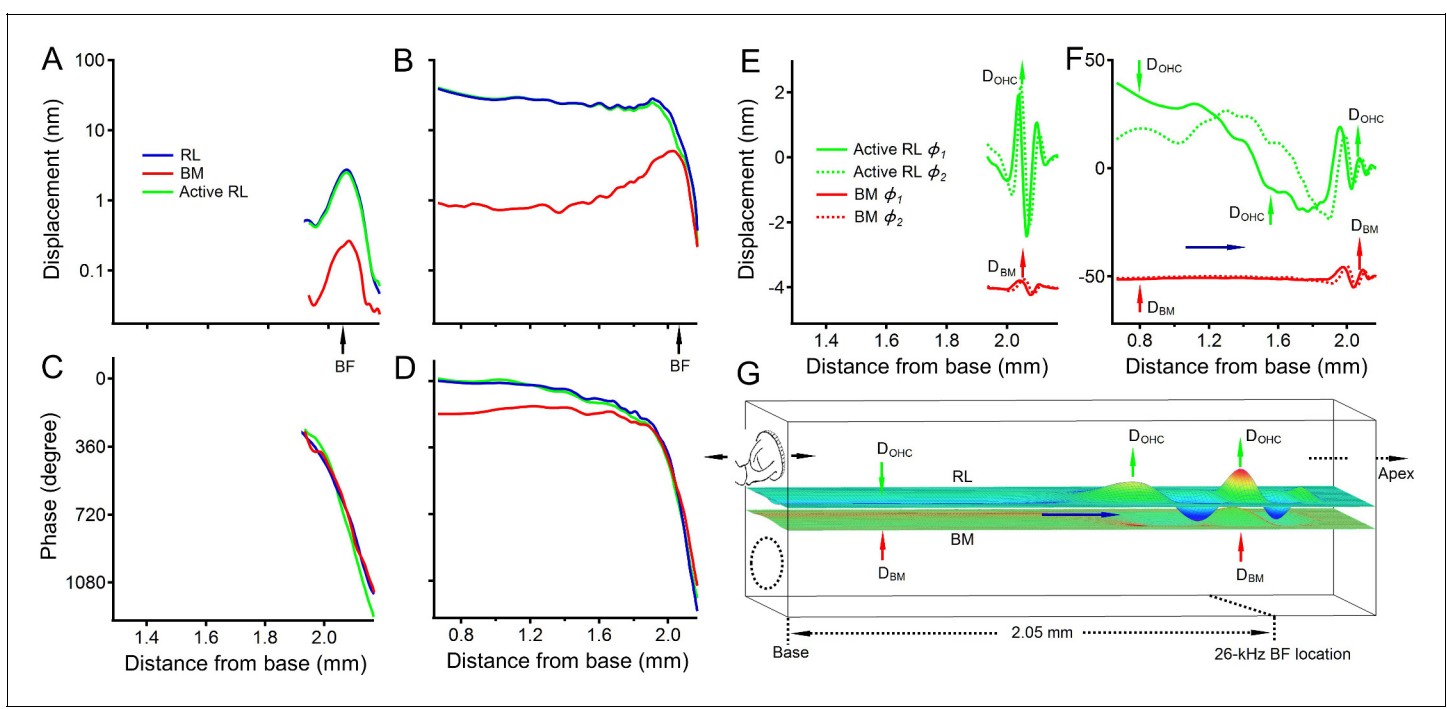

**Figure 5.** Longitudinal patterns of reticular lamina and basilar membrane vibrations. (**A**) The longitudinal patterns of the reticular lamina (RL) and basilar membrane (BM) vibrations at 30 dB SPL. RL and BM vibrations occurred within a 0.2 mm region centered at the best-frequency location. (**B**) At 70 dB SPL, both RL and BM vibrations extended from the best-frequency location to the cochlear base. (**C, D**) RL and BM phase as a function of the longitudinal location at 30 and 70 dB SPL. (**E, F**) Time waveforms of the BM (red) and outer hair cell-driven RL movement (green). For clearer comparison, BM time waveforms were shifted down by 4 nm in panel E and by 50 nm in panel F. The time difference between solid and dotted curves is ~6 µs, equivalent to ~57 degree phase difference at 26 kHz. The outer hair cell-driven active RL vibration (green lines) was obtained by vector subtraction of the BM vibration from the measured RL vibration. $D_{BM}$: basilar membrane displacement; $D_{OHC}$: outer hair cell-driven reticular lamina displacement. (**G**) Diagrams of time waveforms of the BM and RL vibration at an intermediate sound pressure level at 26 kHz.
DOI: https://doi.org/10.7554/eLife.37625.007

at 0.8 mm in *Figure 5F*), depolarized outer hair cells shorten and induce a large downward reticular lamina displacement (green arrow at 0.8 mm in *Figure 5F*). While this movement creates a positive fluid pressure between the reticular lamina and basilar membrane at the cochlear base (0.7–1.2 mm in *Figure 5F,G*), the reticular lamina at locations ~ 1.2–1.8 mm, however, moves upward (green arrows near 1.6 mm in *Figure 5F*), resulting in a negative fluid pressure inside the cochlear partition. This pressure gradient from the base to an apical location likely results in fluid movement in the apical direction inside the cochlear partition (blue arrows in *Figure 5F,G*) as demonstrated in vitro by electrically stimulating the organ of Corti (*Karavitaki and Mountain, 2007*; *Zagadou and Mountain, 2012*). Since the organ of Corti sits on the basilar membrane, the longitudinal fluid movement can travel forward to the best-frequency location as a result of the basilar membrane traveling wave. Thus, a large population of outer hair cells from a broad cochlear area can change the fluid space between the reticular lamina and basilar membrane at the best frequency location on a cycle-by-cycle basis, consequently enhancing the reticular lamina vibration. In addition, in-phase vibrations of the reticular lamina and basilar membrane result in constructive interference near the best-frequency location (*Figure 5E,F,G*), which further enhances reticular lamina vibration at the apical end of outer hair cells. The magnitude of the resulting constructive interference decreases as phase differences move into destructive interference regimes at frequencies below and above the best frequency. While this frequency dependent interference may consequently enhance the tuning of the reticular lamina vibration, its effects probably is relatively small due to the large magnitude difference between the reticular lamina and basilar membrane vibration. The interaction between the reticular lamina and basilar membrane at low frequencies may also be involved in two-tone suppression of the auditory nerve or basilar membrane response (*Delgutte, 1990*; *Ruggero et al., 1992*). It has been shown that the proposed global hydromechanical mechansm is consistent with the observation that auditory nerve activities can be suppressed by stimulating medial olivocochlear efferents or by a low-frequency bias tone not only at the best frequency but also at tail (low) frequencies (*Nam and Guinan, 2017*; *Stankovic and Guinan, 1999*).

Since the stereocilia bundles of both the inner and outer hair cells are anchored in the hair cell cuticular plates which make up a portion of the reticular lamina, the outer hair cell-driven reticular lamina vibration likely results in fluid movement in the subtectorial space and consequently stimulates inner hair cells. It has been shown in vitro that electrical stimulation of outer hair cells of guinea pig cochleae resulted in a counterphasic motion of the tectorial membrane and inner hair cells at frequencies below 3 kHz (*Nowotny and Gummer, 2006*). This result was believed to indicate direct fluid coupling between outer hair cells and inner hair cells through a pulsatile fluid motion. It has also been demonstrated in vitro that outer hair cell stereocilia not only move sideways but also change length in response to sound stimulation (*Hakizimana et al., 2012*). The large bundle deflection was observed when the length change was small, indicating that hair cells are maximally stimulated when the stereocilia length change is minimal. Considering the firm connection of the tallest stereocilia to the tectorial membrane, the stereocilia length change also suggests the interaction between the reticular lamina vibration and the tectorial membrane vibration. Thus, outer hair cells may also play a role in the traveling wave on the tectorial membrane, which has been demonstrated in vitro (*Ghaffari et al., 2007*; *Ghaffari et al., 2010*) and in vivo (*Dong and Cooper, 2006*; *Lee et al., 2015*; *Lee et al., 2016*; *Recio-Spinoso and Oghalai, 2017*; *Rhode and Cooper, 1996*). The interaction between the reticular lamina and tectorial membrane vibration may enhance the stimulus to the inner hair cells and boost hearing sensitivity. Specific mechanisms on how the outer hair cell-driven reticular lamina vibration stimulates inner hair cells, however, remain to be determined experimentally until in vivo micromechanical measurements with cellular spatial resolution become available.

The observed additional delay of the reticular lamina vibration was likely caused by mechanoelectrical (*Corey and Hudspeth, 1979*) and electromechanical (*Brownell et al., 1985*) transduction of outer hair cells and mechanical coupling inside the cochlear partition. Although delays for prestin-associated currents in vitro (*Santos-Sacchi and Tan, 2018*) are longer than the latency difference between the reticular lamina and basilar membrane vibration in vivo, they vary with the membrane voltage of outer hair cells. Voltage excitation away from the resting membrane potential has been shown to have a faster response. Moreover, the depolarized resting potential can minimize the outer hair cell time constant and expanding the bandwidth of the membrane filter by activating voltage-dependent K + conductance (*Johnson et al., 2011*). Mechanical loads on the outer hair cells, such

as in vivo condition, could further improve the cell's frequency response (*Iwasa, 2017*). Since Deiters' cells are located between the outer hair cells and the basilar membrane, the acoustical and cellular forces to and from the outer hair cells have to be transmitted through Deiters' cells. The soft Deiter's cell soma likely induces delays, which can contribute to the delay difference between the reticular lamina and basilar membrane vibration. This apparently is supported by the postmortem magnitude and phase difference between the reticular lamina and basilar membrane vibration (*Figure 2*). When the outer hair cell-generated force is absent under postmortem conditions, the reticular lamina should move passively following the basilar membrane travelling wave with equal magnitude and phase, if the connection between the two structures is rigid. The larger postmortem magnitude and phase difference in gerbils (*Figure 2*) than those in mice (*Figure 2G and H*, *Ren et al., 2016b*) indicate that the mechanical coupling between the reticular lamina and basilar membrane in the gerbil may not be as tight as that in the mouse.

Compared to a large time period ($T$) at a low frequency ($f$) ($T = 1/f$) (such as $T = 2,000$ µs, where $f = 500$ Hz), a 17.9-µs latency difference (*Figure 3J*) is negligible and results in an insignificant phase difference. Thus, the ~180 degree phase difference between the reticular lamina and basilar membrane vibration at a low frequency mainly reflects the opposite movements of both ends of outer hair cells (*Brownell et al., 1985*). The same latency difference, however, can result in ~180 degree phase difference at the best frequency, due to the small period (such as $T = 38$ µs, where $f = 26$ kHz). This latency-induced phase lag of the reticular lamina vibration, compensating for the ~180 degree phase difference observed at low frequencies, accounts for in-phase vibrations of the reticular lamina and basilar membrane at the best frequency. Therefore, the present in vivo results do not conflict with the in vitro observation that both ends of the cylindrical outer hair cells move in opposite directions at low frequencies (*Brownell et al., 1985*; *Santos-Sacchi, 1989*).

In summary, heterodyne low-coherence interferometry demonstrates in vivo that the outer hair cell-driven reticular lamina vibration occurs after, not before, the basilar membrane vibration. The reticular lamina and basilar membrane move in opposite directions at low frequencies and in phase near the best frequency. This experimental finding conflicts with commonly accepted cochlear local feedback theory and suggests that outer hair cells enhance hearing sensitivity through a global hydromechanical mechanism.

## Materials and methods

Twenty-three young healthy Mongolian gerbils of both sexes at age of 4 to 8 weeks (40–80 g) were used in this study.

### Heterodyne low-coherence interferometer

A scanning low-coherence heterodyne interferometer was built based on a scanning laser heterodyne interferometer (*Ren, 2002*; *Ren, 2004*; *Ren et al., 2011*) by replacing the helium-neon laser with a modified superluminescent diode with related optical and electronic components (*Ren et al., 2016a*; *Ren et al., 2016b*). Because of the small coherence length, the low-coherence interferometer can measure vibrations with a high axial resolution (*Chen et al., 2011*; *Hong and Freeman, 2006*; *Lee et al., 2015*). The use of low-coherence light and an objective lens with numerical aperture 0.42 provides adequate spatial selectivity for measuring the reticular lamina and basilar membrane vibrations in the living cochlea. In addition to its unprecedented sensitivity, this interferometer has a broad dynamic range, high temporal resolution, and low phase noise, due to the use of a 40-MHz carrier for heterodyne detection. In contrast to homodyne interferometers, a heterodyne interferometer can detect vibration directions without 180-degree phase uncertainty (*Hong and Freeman, 2006*; *Khanna et al., 1986*; *Lukashkin et al., 2005*), which ensured the reliability of the phase measurements in this study.

### Measurement of reticular lamina and basilar membrane vibrations

Animal anesthesia and surgical procedures were the same as described previously (*Ren, 2002*; *Ren et al., 2011*; *Ren and Nuttall, 2001*). Briefly, after about one third of the round window membrane was removed and the opened round window was partially covered with a glass coverslip, the object light from the interferometer was focused on the center of the outer hair cell region of the cochlear partition at the basal turn (*Figure 6A*). The transverse locations of the basilar membrane

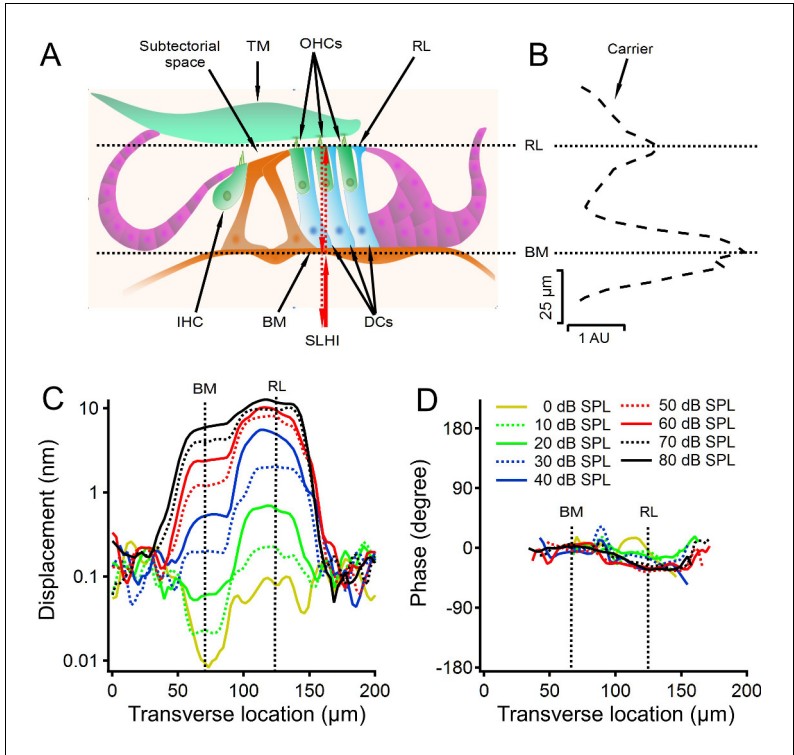

**Figure 6.** Diagram of a cross section of the organ of Corti and the cochlear partition vibration as a function of the transverse location. (**A**) Diagram of a cross section of the organ of Corti. TM, tectorial membrane; OHCs, outer hair cells; RL, reticular lamina; IHC, inner hair cell; BM, basilar membrane; DCs, Deiters' cells. SLHI, scanning low-coherence heterodyne interferometer. Red arrowed lines indicate incident and backscattered low-coherence lights. (**B**) The backscattered light level (carrier) as a function of the transverse location shows two peaks at locations on the BM and RL. (**C**) Displacement as a function of the transverse location at different sound levels. Displacements at the RL location are greater and show more compressive growth with sound level than those at the BM location. Line types and colors in this plot are the same as those in panel D. (**D**) Phase decreased slightly at the RL location. Data were collected from a sensitive gerbil cochlea from the basal turn with best frequency 30 kHz. Due to the extremely low reflectivity of perilymph, the noise level outside the cochlear partition is higher than low-level tone-induced vibrations at the RL and BM location in panel C.

DOI: https://doi.org/10.7554/eLife.37625.008

and reticular lamina were determined by measuring the backscattered light (carrier) level as a function of the transverse location. The locations of the basilar membrane and reticular lamina were indicated by the two peaks of the backscattered light level (*Figure 6B*) and confirmed by the distinct magnitude and phase of the cochlear partition vibrations at the two locations (*Figure 6C,D*). Cochlear partition vibrations were measured as a function of the transverse location at the best frequency (30 kHz) and at different sound levels (0–80 dB SPL).

When the object beam of the interferometer was focused on the basilar membrane or the reticular lamina, acoustical tones at different frequencies and levels were delivered to the ear canal. The tone frequency was changed from 1.8 to 40.0 kHz by ~0.2 kHz per step. The magnitude and phase of the cochlear partition vibration were measured using a lock-in amplifier (SR830, Stanford Research System, Inc. Sunnyvale, CA) and recorded on a computer. The best frequency was determined by the peak of the basilar membrane displacement as a function of frequency at 30 dB SPL. For recording time waveforms of the reticular lamina and basilar membrane vibration, a 10-µs electrical pulse was generated by a dynamic signal analyzer (PXI-4461, National Instruments, Austin, TX) and used to drive an electrostatic speaker (EC1, Tucker-Davis Technologies, Alachua, FL). Displacements of the reticular lamina and basilar membrane vibrations were digitized at the rate of 200,000 samples per second and averaged synchronously with stimuli for 100 times. Time waveforms from the same time window were plotted to show the temporal relationship between the reticular lamina and basilar

membrane vibration (*Figure 4*). The reticular lamina and basilar membrane vibration were measured in a random order.

Cochlear sensitivity was monitored by continuously recording distortion product otoacoustic emission (DPOAE). The DPOAE at 16 kHz was evoked by two 60 dB SPL tones at 20 and 24 kHz. A cochlea with <5 dB DPOAE decrease was considered sensitive. Postmortem data were collected 10 to 30 min after the animal's death from anesthetic overdose. Stapes vibration was recorded under the same conditions as for the cochlear mechanical measurement.

### Data analysis and statistics

Igor Pro (Version 7.0.5.2, WaveMetrics, Lake Oswego, OR) was used for analyzing data. The frequency responses of the reticular lamina and basilar membrane were presented by displacement and phase as a function of frequency. The transfer functions were estimated by the displacement ratio of the reticular lamina or the basilar membrane to the stapes at different frequencies. Since the phase lag of the cochlear partition vibration ($\phi$) is a function of latency ($\tau$) ($\phi = 2\pi f\tau$, where $f$ is frequency), the time relationship between the basilar membrane and reticular lamina vibration were presented by the phase difference ($\Delta\phi$) ($\Delta\phi = \phi_{RL} - \phi_{BM}$), where $\phi_{RL}$ is the reticular lamina phase and $\phi_{BM}$ is the basilar membrane phase at different frequencies. The latency difference ($\Delta\tau$) was derived from the slope of the linear regression line of the phase difference as a function of frequency ($\Delta\phi/\Delta f$) ($\Delta\tau = \Delta\phi/2\pi\Delta f$).

The latency difference ($\Delta\tau$) was also determined based on the time waveform of the reticular lamina and basilar membrane response to clicks. Despite the sharp onset of the electrical pulse, the onsets of the reticular lamina and basilar membrane response were distorted due to frequency bandwidth limits of the speaker, the middle ear, and the cochlea, and cannot been measured precisely. Because the first displacement peak of the time waveform indicates the arriving time of the maximum stimulation to the reticular lamina and the basilar membrane, $\Delta\tau$ was determined by the time difference between the first peak of the reticular lamina ($T_B$) and that of the basilar membrane ($T_A$), ($\Delta\tau = T_B - T_A$) (*Figure 4C*). The longitudinal patterns of the reticular lamina and basilar membrane vibrations were presented by plotting displacement and phase as a function of the longitudinal location. The longitudinal location was derived from the stimulus frequency according to the frequency-location function in gerbil cochleae (*Müller, 1996*). The grouped results were presented by mean and standard error calculated across animals. Sound level-dependent latency changes were tested at 50, 60, 70, and 80 dB SPL using one-way ANOVA and $p < 0.05$ was considered significantly different.

## Acknowledgements

We thank John V Brigande for critical comments on the manuscript, Alfred L Nuttall and other colleagues at Oregon Hearing Research Center for helpful discussion of the data, Edward Porsov for engineering support, and Santos-Sacchi and William Brownell for their helpful guidance in revising this paper. This study was funded by NIH grants R01 DC004554 (TR).

## Additional information

### Funding

| Funder | Grant reference number | Author |
| --- | --- | --- |
| National Institutes of Health | R01-DC004554 | Tianying Ren |

The funders had no role in study design, data collection and interpretation, or the decision to submit the work for publication.

### Author contributions

Wenxuan He, Conceptualization, Data curation, Formal analysis, Validation, Investigation, Visualization, Methodology, Writing—review and editing; David Kemp, Formal analysis, Validation, Writing—review and editing; Tianying Ren, Conceptualization, Data curation, Formal analysis, Supervision,

Funding acquisition, Validation, Investigation, Visualization, Methodology, Writing—original draft, Project administration, Writing—review and editing

## Author ORCIDs
Tianying Ren  http://orcid.org/0000-0002-2533-7203

## Ethics
Animal experimentation: All experiments and procedures were performed according to protocols approved by the Animal Care and Use Committee of the National Institute on Deafness and Other Communication Disorders and the Animal Care and Use Committee of the Oregon Health & Science University (Protocol Number: IP00000932).

## Decision letter and Author response
Decision letter https://doi.org/10.7554/eLife.37625.011
Author response https://doi.org/10.7554/eLife.37625.012

## Additional files

### Supplementary files
• Transparent reporting form
DOI: https://doi.org/10.7554/eLife.37625.009

### Data availability
All data generated or analysed during this study are included in the manuscript and supporting files.

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
