## [Decision Letter]

Thank you for submitting your article "Timing of the reticular lamina and basilar membrane vibration in living gerbil cochleae" for consideration by *eLife*. Your article has been reviewed by Andrew King as the Senior Editor, a Reviewing Editor, and three reviewers. The following individuals involved in review of your submission have agreed to reveal their identity: Joseph Santos-Sacchi (Reviewer #1); William Brownell (Reviewer #3).

The reviewers have discussed the reviews with one another and the Reviewing Editor has drafted this decision to help you prepare a revised submission.

Summary:

The study advances our understanding of mammalian hearing by measuring the timing of two structures within the cochlear partition. Hearing scientists have studied basilar membrane mechanics for more than 70 years and recently, with advances in optical tools, the mechanics of the reticular lamina, which forms the upper surface of the organ of Corti. Previous studies established that outer hair cell electromotility causes these two structures to move in opposite directions and that reticular laminar deflections are an order of magnitude greater than those of the basilar membrane. Using laser interferometry, the authors observed a phase difference between the motion of these two structures at different sound frequencies and inferred from this that the reticular lamina lags the basilar membrane vibration by 20-30 microseconds. A similar delay between the two parts of the organ of Corti was observed by stimulating the cochlea with broad-band clicks. Although comparable phase behavior has previously been reported by the authors in mice, the present study extends the phase analysis to the time domain and reveals the reticular lamina responds between 20-30 microseconds after the basilar membrane. This finding has negative implications for the notion that outer hair cells amplify the response of the basilar membrane by providing local, cycle-by-cycle feedback, and, together, the two studies highlight the possibility that the degree of coupling between outer hair cells and surrounding structures may be critical in determining the mechanical response of the cochlea. The reviewers stated that the data are robust and that this is a significant finding but felt that a more explicit discussion of the insights provided by this work into the nature of cochlear mechanics is needed.

Essential revisions:

The current Discussion section addresses the consequences of the phase difference between the reticular lamina and the basilar membrane at low frequencies for longitudinal fluid movement. This has been covered before in the authors' previous paper in which mice were used and should be largely removed. They should instead focus on the implications of the current results for the nature of the mechanical interactions that take place within the cochlea. In doing so, the reviewers recommended the following factors that should be considered.

1) The present study proposes a "global hydromechanical" mechanism to account for the reticular lamina response delay and broadening of its frequency response. The authors' 2016 paper introduced the mechanism. It posits that outer hair cell length changes result in local changes in the volume of fluid-filled spaces between the reticular lamina and basilar membrane. Fluid movement into or away from the region of interest will introduce delays in the response of the reticular lamina. The mechanism is plausible, particularly for the response to intense stimuli. However, additional hydrodynamic and viscous damping effects may be associated with other structures in the cochlear partition. The manuscript should include a brief discussion of these mechanisms. The stereocilia bundle of both the inner and outer hair cells is anchored in the hair cell cuticular plate that makes up a portion of the reticular lamina. The tallest row of OHC stereocilia are firmly connected to the tectorial membrane. The stereocilia bundle has been shown to undergo length changes in vitro and fluid movement in the subtectorial space has long been postulated to mechanically stimulate inner hair cells. This movement will augment the forces acting on the reticular lamina. In addition, in vitro and in vivo vibrations of the tectorial membrane result in longitudinal and radial traveling wave behavior. Thus, tectorial membrane dynamics may further augment those arising from the mechanism proposed by the authors.

2) Both the basilar membrane and reticular lamina move in phase near the best frequency. The magnitude of the resulting constructive interference decreases as phase differences move into destructive interference regimes at higher and lower frequencies. The low frequency region may be associated with two-tone suppression regions observed in the response area of single auditory neurons. The implications of this should be discussed briefly.

3) Further discussion is needed of potential species differences in mechanical coupling within the cochlea and whether the phase data might reflect coupling between the basilar membrane and reticular lamina via the soft Deiters' cell soma.

At what frequency were the measurements shown in Figure 6? Why is there no phase difference between the basilar membrane and the reticular lamina? It would also be helpful to know which colors correspond to which SPLs.

Because in vitro and in vivo electrical stimulation experiments have shown that outer hair cells can induce basilar membrane motion, is there an intrinsic error is assuming that the difference between basilar membrane and reticular laminar movements exactly defines outer hair cell contributions?

---

## [Author Response]

Essential revisions:The current Discussion section addresses the consequences of the phase difference between the reticular lamina and the basilar membrane at low frequencies for longitudinal fluid movement. This has been covered before in the authors' previous paper in which mice were used and should be largely removed. They should instead focus on the implications of the current results for the nature of the mechanical interactions that take place within the cochlea. In doing so, the reviewers recommended the following factors that should be considered.

According to the reviewers' suggestions, the original description of the global mechanical mechanism has been shortened significantly, and the Discussion section has been changed to focus on the implication of the current results on mechanical interactions within the cochlear partition.

1) The present study proposes a "global hydromechanical" mechanism to account for the reticular lamina response delay and broadening of its frequency response. The authors' 2016 paper introduced the mechanism. It posits that outer hair cell length changes result in local changes in the volume of fluid-filled spaces between the reticular lamina and basilar membrane. Fluid movement into or away from the region of interest will introduce delays in the response of the reticular lamina. The mechanism is plausible, particularly for the response to intense stimuli. However, additional hydrodynamic and viscous damping effects may be associated with other structures in the cochlear partition. The manuscript should include a brief discussion of these mechanisms. The stereocilia bundle of both the inner and outer hair cells is anchored in the hair cell cuticular plate that makes up a portion of the reticular lamina. The tallest row of OHC stereocilia are firmly connected to the tectorial membrane. The stereocilia bundle has been shown to undergo length changes in vitro and fluid movement in the subtectorial space has long been postulated to mechanically stimulate inner hair cells. This movement will augment the forces acting on the reticular lamina. In addition, in vitro and in vivo vibrations of the tectorial membrane result in longitudinal and radial traveling wave behavior. Thus, tectorial membrane dynamics may further augment those arising from the mechanism proposed by the authors.

In addition to the interaction between the reticular lamina and basilar membrane vibration, effects of the outer hair-driven reticular lamina vibration on other structures in the cochlear partition are discussed by adding the following paragraph to the Discussion section.

"Since the stereocilia bundles of both the inner and outer hair cells are anchored in the hair cell cuticular plates which make up a portion of the reticular lamina, the outer hair cell-driven reticular lamina vibration likely results in fluid movement in the subtectorial space and consequently stimulates inner hair cells. […] Specific mechanisms on how the outer hair cell-driven reticular lamina vibration stimulates inner hair cells, however, remain to be determined experimentally until in vivo micromechanical measurements with cellular spatial resolution become available."

2) Both the basilar membrane and reticular lamina move in phase near the best frequency. The magnitude of the resulting constructive interference decreases as phase differences move into destructive interference regimes at higher and lower frequencies. The low frequency region may be associated with two-tone suppression regions observed in the response area of single auditory neurons. The implications of this should be discussed briefly.

The frequency dependent interference between the reticular lamina and basilar membrane vibration has been addressed in the following text (Discussion section).

"The magnitude of the resulting constructive interference decreases as phase differences move into destructive interference regimes at frequencies below and above the best frequency. […] The interaction between the reticular lamina and basilar membrane at low frequencies may also be involved in two-tone suppression of the auditory nerve or basilar membrane response (Delgutte, 1990; Ruggero et al., 1992)."

3) Further discussion is needed of potential species differences in mechanical coupling within the cochlea and whether the phase data might reflect coupling between the basilar membrane and reticular lamina via the soft Deiters' cell soma.

We agree with the reviewers that the phase data might partially reflect coupling between the basilar membrane and reticular lamina via the soft Deiters' cell soma. The species difference in mechanical coupling within the cochlea has been discussed in the following paragraph, which has been included in the revised manuscript (Discussion section).

"Since Deiters' cells are located between the outer hair cells and the basilar membrane, the acoustical and cellular forces to and from the outer hair cells have to be transmitted through Deiters' cells. […] The larger postmortem magnitude and phase difference in gerbils (Figure 2) than those in mice (Figure 2G and H, Ren et al., 2016b) indicate that the mechanical coupling between the reticular lamina and basilar membrane in the gerbil may not be as tight as that in the mouse."

At what frequency were the measurements shown in Figure 6? Why is there no phase difference between the basilar membrane and the reticular lamina? It would also be helpful to know which colors correspond to which SPLs.

The measurements shown in Figure 6 were taken at 30 kHz, the best frequency of the measured location. This information has been included in the text (subsection “Measurement of reticular lamina and basilar membrane vibrations”) and the legend of Figure 6. The reason for no phase difference between the basilar membrane and reticular lamina is that the vibrations were measured at the best frequency. This is consistent with the data shown in Figures 1H and 3J. Line legend has been included in Figure 6D to show responses at different sound pressure levels.

Because in vitro and in vivo electrical stimulation experiments have shown that outer hair cells can induce basilar membrane motion, is there an intrinsic error is assuming that the difference between basilar membrane and reticular laminar movements exactly defines outer hair cell contributions?

It is true that outer hair cells can induce basilar membrane motion in vitro and in vivo. This, however, may not necessarily conflict with the assumption that the outer hair cell contributions can be estimated by the difference between the basilar membrane and reticular lamina vibration, because the two structures move at different times due to the latency difference. We are aware of the limitation of our method for estimating the outer hair cell contribution due to the lack of information on mechanical properties of Deiters' cells. Our statement has been softened by changing "obtained" into "estimated" in the Discussion section.